# Sulfo-Gambierones, Two New Analogs of Gambierone Produced by *Gambierdiscus excentricus*

**DOI:** 10.3390/md19120657

**Published:** 2021-11-24

**Authors:** Thomas Yon, Manoëlla Sibat, Elise Robert, Korian Lhaute, William C. Holland, R. Wayne Litaker, Samuel Bertrand, Philipp Hess, Damien Réveillon

**Affiliations:** 1Laboratoire Phycotoxines, DYNECO, Ifremer, F-44000 Nantes, France; manoella.sibat@ifremer.fr (M.S.); Elise.Robert@ifremer.fr (E.R.); Korian.Lhaute@ifremer.fr (K.L.); damien.reveillon@ifremer.fr (D.R.); 2Beaufort Laboratory, National Centers for Coastal Ocean Science, National Ocean Service, NOAA, Beaufort, NC 28516, USA; chris.holland@noaa.gov; 3CSS, Inc. Under Contract to National Oceanic and Atmospheric Administration, National Centers for Coastal Ocean Science, National Ocean Service, Beaufort, NC 28516, USA; wayne.r.litaker@noaa.gov; 4Institut des Substances et Organismes de la Mer, ISOMER, Nantes Université, UR 2160, F-44000 Nantes, France; samuel.bertrand@univ-nantes.fr; 5ThalassOMICS Metabolomics Facility, Plateforme Corsaire, Biogenouest, F-44311 Nantes, France

**Keywords:** *Gambierdiscus excentricus*, sulfo-gambierones, high-resolution mass spectrometry, Neuro2a cell-based assay

## Abstract

Ciguatera poisoning is caused by the ingestion of fish or shellfish contaminated with ciguatoxins produced by dinoflagellate species belonging to the genera *Gambierdiscus* and *Fukuyoa*. Unlike in the Pacific region, the species producing ciguatoxins in the Atlantic Ocean have yet to be definitely identified, though some ciguatoxins responsible for ciguatera have been reported from fish. Previous studies investigating the ciguatoxin-like toxicity of Atlantic *Gambierdiscus* species using Neuro2a cell-based assay identified *G. excentricus* as a potential toxin producer. To more rigorously characterize the toxin profile produced by this species, a purified extract from 124 million cells was prepared and partial characterization by high-resolution mass spectrometry was performed. The analysis revealed two new analogs of the polyether gambierone: sulfo-gambierone and dihydro-sulfo-gambierone. Algal ciguatoxins were not identified. The very low ciguatoxin-like toxicity of the two new analogs obtained by the Neuro2a cell-based assay suggests they are not responsible for the relatively high toxicity previously observed when using fractionated *G. excentricus* extracts, and are unlikely the cause of ciguatera in the region. These compounds, however, can be useful as biomarkers of the presence of *G. excentricus* due to their sensitive detection by mass spectrometry.

## 1. Introduction

Species of benthic dinoflagellates in the genera *Gambierdiscus* and *Fukuyoa* have been widely studied because of their role in producing neurotoxins that can bioaccumulate in the food chain, in particular the potent group of ciguatoxins [1,2]. Consumption of fish or shellfish contaminated with such toxins can cause ciguatera poisoning (CP), characterized by gastrointestinal, cardiological and mild to severe and long-lasting neurological symptoms [3,4]. This illness is widespread in tropical areas where these dinoflagellates occur, with an estimated 10,000 to 50,000 intoxications occurring annually [1]. The first conclusive evidence that ciguatoxins were produced by benthic dinoflagellates resulted in the description of a new genus and species—*Gambierdiscus toxicus* [5,6]. For almost two decades, the genus was assumed to contain a single species, which was responsible for all CP events. Following a study published in 1999, a series of ever-increasing taxonomic investigations have resulted in the division of the original genus into two separate genera, “*Gambierdiscus*” and “*Fukuyoa*”, containing 18 and 3 described species, respectively [3,7,8,9]. Initially, two major types of toxins potentially involved in ciguatera poisoning were discovered in fish flesh: maitotoxins (MTX), which are large non-polymeric amphiphilic polyethers, initially found in the viscera of the bristletooth surgeonfish *Ctenochaetus striatus* [10,11], and ciguatoxins (CTX), which are more apolar polyethers, firstly found in moray eel, *Gymnothorax javanicus* [12].

After in-depth risk evaluation and many toxicological studies, it appears that ciguatoxins are more likely the cause of CP rather than maitotoxins. Ciguatoxins have been demonstrated to bio-accumulate through the food chain [13]. They have been successfully isolated from fish tissue and proven to be voltage-gated sodium channel activators and voltage-gated potassium channels blockers, whose mechanism of action accounts for the diverse range of observed symptoms associated with ciguatera poisoning [4,14,15,16]. In contrast, even though maitotoxins have been shown to increase calcium ion influx through excitable membranes, resulting in cell depolarization, hormone and neurotransmitter secretion, and breakdown of phosphoinositides, they are water soluble and do not readily accumulate in the food chain [17]. As maitotoxins have primarily been isolated from fish liver tissues, with only trace amounts present in the flesh [18], they are only considered potentially significant in cases when fish livers are being consumed, as may be the case in some highly CP endemic regions.

To date, only the species *G. polynesiensis* (previously identified as *G. toxicus* [8]) was demonstrated by analog-specific methods, such as liquid chromatography coupled to tandem or high-resolution mass spectrometry (LC-MS/MS or LC-HRMS), as a ciguatoxin producer in the Pacific Ocean [19] with more than 14 ciguatoxin analogs identified [20]. The biotransformation of ciguatoxins produced by this species as they passed through the food chain from dinoflagellates to fish has been elucidated [21,22,23]. In the Atlantic Ocean, a different structure of ciguatoxin was found in horse-eye jack, *Caranx latus* [24,25], however, so far, no algal precursor has been identified and thus no *Gambierdiscus* species producing such precursors has been reported.

Other polyethers produced by species of the genus *Gambierdiscus* were discovered in algal cultures (mainly as a by-product when screening for ciguatoxin- or maitotoxin-type activity) [26]. Such compounds include gambieric acid-A to -D [27], also showing antifungal activities; gambierol [26], which acts as a blocker of voltage-gated potassium channels [28]; as well as gambieroxide [29] or gambierone [30] and its analogue 44-methylgambierone (previously reported as MTX3, but later shown to have a structure and activity more closely related to that of gambierone [31,32]). Gambierone and 44-methylgambierone did show ciguatoxin-like activity, but at levels 200 or 55,500 times lower than ciguatoxin (P-CTX3C) based on neuroblastoma cell (Neuro2a) assay [31] and intraperitoneal injection in mice [33,34], respectively.

In contrast to ciguatoxins and maitotoxins, which were reported in only few strains of *Gambierdiscus*, gambierone and/or 44-methylgambierone were demonstrated to be produced by 16 out of the 18 species of *Gambierdiscus* (all except *G. jejuensi* and *G. excentricus* [35]), by two species of *Fukuyoa* (*F. paulensis* and *F. ruetzeri*), and by three species of *Coolia* (*C. canariensis*, *C. malayensis*, *C. tropicalis*) [32,33,36].

Since 2011, in vitro toxicity studies searching for ciguatoxin-like activity carried out mainly using the Neuro2a cell-based-assay, suggested *G. excentricus* as one of the most toxic species among all *Gambierdiscus* species encountered in the Atlantic Ocean [37,38,39,40]. The effort of toxin characterization performed on *Gambierdiscus excentricus* demonstrated this species as a maitotoxin producer with the presence of maitotoxin 4 (MTX4) [35,41]. Interestingly, MTX4 seems restricted to this species, as it was only detected in all 7 strains of *G. excentricus,* but not in the other 37 strains (corresponding to 13 species) of *Gambierdiscus* and *Fukuyoa* screened so far [35,41]. The toxin profile of *G. excentricus* was explored in the same study and two other putative compounds were reported: a putative MTX2 with the transition *m/z* 1091.5/96.9 in low-resolution mass spectrometry using multiple reaction monitoring mode (MRM) (based on [M-3H]^3−^/[HOSO_3_]^−^ of MTX2 [42]), which turned out to be a mono-charged ion when examined by high-resolution mass spectrometry, and a second compound, with the MRM transition *m/z* 1037.5/96.8, which would correspond to the transition [M-H]^−^/[HOSO_3_]^−^ of 44-methylgambierone but had a different retention time in all the strains of *G. excentricus* compared to 44-methylgambierone in the 44 strains of other species. Analysis performed with high-resolution mass spectrometry revealed the presence of *m/z* 1037.5820 that corresponded to a mass error of 99 ppm compared to the theoretical mass of 44-methylgambierone (*m/z* 1037.4785 for [C_52_H_77_O_19_S]^−^). Thus, it was unlikely that this compound was an isomer of 44-methylgambierone.

This study further investigated the metabolites of *G. excentricus* using low and high-resolution mass spectrometry and provides new insights into the toxin profile of this species. In particular, we report two sulfated analogs of gambierone, and confirm the apparent absence of their toxicity at an estimated concentration of >500 ng mL^−1^ in Neuro2a cells.

## 2. Results and Discussion

### 2.1. Revisiting the Toxin Profile of G. excentricus

In the present study, the signals were first obtained using low-resolution mass spectrometry in negative ionization electrospray (ESI^−^) (MRM mode focusing on maitotoxins, gambierones, gambieric acids and gambieroxide (see Section 3.7, method 1)). Signals integrated corresponding to compounds responding to at least one transition (Appendix A), regardless of the retention time, are presented in Figure 1A and Table 1. This approach allows putative analogs of known compounds or simply isobaric compounds to be detected. The profile obtained was consistent with the one reported by Pisapia et al. [35] (Table 1 in blue), with the peak corresponding to MTX4 at 5.99 min (Figure 1A (**5**); hereafter, the bolded numbers in parentheses indicate specific peaks in the referenced figure indicating the presence of a unique compound), a peak at 6.99 min with one MRM transition of MTX2 (Figure 1A, (**6**)) and a third peak at 5.03 min with two MRM transitions of 44-methylgambierone (Figure 1A, (**3**)). However, with the new availability of gambierone and 44-methylgambierone standards and the implementation of more transitions in the MRM method, other putative compounds were observed. Namely, two putative gambierones at 4.47 min and 7.08 min (Figure 1A (**1**) and (**7**)), a putative MTX1 eluting at 4.99 min (Figure 1A, (**2**)), a putative gambieric acid at 5.72 min (Figure 1A (**4**)) and a putative 44-methylgambierone eluting at 7.82 min (Figure 1A (**8**)). Standards of gambierone, 44-methylgambierone and MTX1 injected for reference eluted at 5.76, 6.01, and 6.09 min, respectively (Figure 1B).

The putative compounds were then examined with high-resolution mass spectrometry in full-scan mode using a method adapted from Sibat et al. [43], able to detect a wide variety of compounds (see Section 3.8, method 2) to confirm or not the putative molecular formula based on the exact monoisotopic mass. The mass-to-charge ratio (*m/z*) values measured by low resolution mass spectrometry (LRMS) for each precursor ion was searched with a tolerance window based on the performance of a quadrupole (i.e., +/−0.7 Da). Results are presented in the Table 1 and in Supplementary Appendix A.

Among the eight peaks corresponding to the transitions monitored in the extract of *G. excentricus* (Table 1), five were not confirmed by HRMS. The compounds did either not correspond to the charge state expected (i.e., for (**2**) see Appendix A and for (**6**) see Appendix A) or were not detected due to the difference of sensitivity of the HRMS for (**7**) (data not shown). The compounds (**4**) and (**8**) (see Appendix A) were only observed in the LRMS system because the intensity of their second or third isotope was higher than the limit of detection.

For the three other signals:MTX4 (Figure 1A, peak **(5)** and Table 1) was confirmed on HRMS with a high exact mass accuracy (−2.9 ppm) (Appendix A) and in accordance with Pisapia et al. [41].A putative 44-methylgambierone (Figure 1A, peak (**3**)), was found on the HRMS system at 6.7 min (Appendix A) with a monoisotopic mass of *m/z* 1037.4599 (Appendix A), which corresponded to a mass error of −17.9 ppm compared to the theoretical mass of the [M-H]^−^ ion of 44-methylgambierone. In positive ionization mode (ESI^+^), the chromatographic peak at 6.7 min did not present any ions with a mass related to the expected [M+H]^+^ at *m/z* 1039.4931 (Appendix A) but the ion *m/z* 1056.5047 could correspond to the ammonium adduct [M+NH_4_]^+^ of 44-methylgambierone, however, with a mass error of −14.1 ppm. In addition, the protonated molecule, sodium adducts and several in-source water losses commonly observed for gambierones, ciguatoxins or related cyclic polyethers [20,45] were not observed for this compound, which also suggests a weak structural resemblance with 44-methylgambierone.Finally, the putative gambierone, compound (**1**) was observed in HRMS at 6.2 min in both ESI^+^ and ESI^−^ modes (Appendix A). Interestingly, the exact mass of the mono-isotopic ion (*m/z* 1023.4596) was close to the one of the [M–H]^−^ of gambierone with a mass error of −3.2 ppm. The same observation was made in ESI^+^ mode with three ions reported as the water loss [M–H_2_O+H]^+^ (i.e., *m/z* 1007.4684, +1.6 ppm), the molecular ion [M+H]^+^ (*m/z* 1025.4849, +7.3 ppm) and the ammonium adduct [M+NH_4_]^+^ (*m/z* 1042.5034, −0.6 ppm) of gambierone.Peak (**1**) eluted 1.30 min earlier than the standard of gambierone (Figure 1A,B) in both systems (method 1 and 2), suggesting it to be a polar analog of gambierone. However, the low intensity of the signal observed in this *G. excentricus* extract (i.e., 250,000 cell mL^−1^), prevented further investigation. Therefore, we purified this compound from a higher biomass, and performed several HRMS and HRMS/MS analyses to better characterize this unknown analog.

### 2.2. Characterization of the Putative Gambierone Analogs by HRMS

The putative gambierone analog, peak (**1**) Figure 1A, was purified from a pool of 124 million cells of two *G. excentricus* strains “Bahamas Gam5” and “Pulley-Ridge Gam2” using a combination of selective extraction, liquid-liquid partitioning, size exclusion chromatography, and preparative high-performance liquid chromatography (preparative-HPLC). Finally, a total of 70.6 µg gambierone-equivalent was obtained from this purification process.

#### 2.2.1. Effect of Ionization Parameters on In-Source Fragmentation and Adduct Formation

The analysis of a fraction obtained with preparative-HPLC (containing approx. 600 ng mL^−1^ gambierone eq. of the putative gambierone) was performed in full scan mode to identify the different ions associated with compound (**1**).

The full scan of the putative gambierone analog (Figure 2A) and the table of mass errors associated (Table 2) demonstrated the presence of two groups of ions starting with the *m/z* 1042.5072 corresponding to the ammonium adduct (+3.1 ppm) of gambierone, followed by the protonated molecule and two successive water losses. The second group corresponded to the loss of SO_3_ followed by four successive water losses. The same profile was reported by Estevez et al. [45], and Rodriguez et al. [30]. The ion detected at *m/z* 962.5498 (+2.7 ppm), not reported in the literature for gambierone, was suggested as an ammonium adduct of the *m/z* 945.5206.

Based only on the full scan profile acquired in ESI^+^ mode with the method 2 (Figure 2A), no difference in structure, mass or in-source behavior could explain the difference in terms of retention time between this analog and gambierone. In our reverse-phase chromatography, the difference in retention time between gambierone and its methylated analog 44-methylgambierone was close to 0.25 min, hence the earlier elution (1.3 min) observed for the putative gambierone analog (i.e., peak (**1**)) could not reasonably be attributed to an isomeric compound, e.g., a different position of a methyl function.

Moreover, in the ESI^−^ mode (Figure 2B) the ion observed at *m/z* 1023.4639 corresponding to the [M-H]^−^ anion of gambierone was not the base peak ion as expected for gambierone, and two other more intense ions were observed at *m/z* 1103.4209 and *m/z* 1125.4023, resulting in an exact mass difference of 79.9570 Da and 101.9384 Da, respectively. The loss of 79.9570 Da is characteristic of a sulfur trioxide loss, as already shown in the ESI^+^ mode (Figure 2A; i.e., from *m/z* 1025.4774 to *m/z* 945.5206), thus suggesting the presence of a second sulfate group in this analog of gambierone. In addition, the presence of a second sulfate group would increase the polarity, therefore explaining the earlier retention time of peak (**1**) compared to gambierone on both systems (methods 1 and 2). In analogy, the ion *m/z* 1125.4023 corresponds to the sodium adduct of *m/z* 1103.4209 with a mass error of +0.6 ppm, and the *m/z* 1103.4209 was attributed to the deprotonated molecule (+1.1 ppm). Altogether, these results and the small Δppm were consistent with the molecular formula C_51_H_76_O_22_S_2_ and the hypothesis of a sulfated analog of gambierone.

In order to reduce in-source fragmentation and to obtain a higher intensity for the [M+H]^+^ of the sulfated gambierone analog, a new method adapted from Yon et al. [20] was implemented; method 3 (See Section 3.8). The resulting full scan in ESI^+^ mode presented in Figure 2C revealed a much greater abundance of sulfated ion (i.e., from *m/z* 1105.4342 to 1069.4131) and its corresponding ammonium and di-ammonium adducts (*m/z* 1122.4608 and 1139.4874, respectively) with high mass accuracy (Table 2). Once the first sulfate group was lost in-source, the ions observed for this analog were the same as the one of gambierone except for *m/z* 962.5472 (+0.5 ppm), which corresponds to the loss of the two sulfate functions associated with an ammonium adduct ([M-2SO_3_+NH_4_]^+^). Thus, we name the peak (**1**): sulfo-gambierone.

#### 2.2.2. Discovery of a Second Analogue

The analysis performed with method 3, on a concentrated pool of preparative-HPLC fractions containing 29.8 µg mL^−1^ gambierone eq. (sum of putative gambierone analogs) in ESI^+^ full scan mode, revealed the presence of a second compound with a delta of +2 Da for all ions (Appendix A). The chromatographic gradient was thus modified to separate the two compounds and acquire separated mass spectra (Figure 3), leading to the creation of method 4 (see Section 3.8).

In addition to their similar retention time on reversed phase chromatography, both compounds were very similar in terms of ion ratios, in-source fragmentation and adduct formation. The delta of 2.015 ± 0.002 Da was present on all detected ions, suggesting the presence of two additional hydrogens on the structure of the second gambierone analog. The molecular formula proposed is thus C_51_H_78_O_22_S_2_ with an error on the exact mass of –0.4 ppm (*m/z* 1124.4760) for the ammonium adduct (Figure 3A(1b),C) and the compound was named dihydro-sulfo-gambierone accordingly.

The gambierone standard, purified sulfo-gambierone (1a), and dihydro-sulfo-gambierone (1b) were analyzed by targeted MS/MS to evaluate the differences in fragmentations between the three compounds under the same analytical conditions.

#### 2.2.3. Fragmentation Pathways of Sulfo-Gambierone

The targeted MS/MS spectra of sulfo-gambierone is presented in Figure 4, and the proposed attribution of ion formula is presented in Appendix A.

The fragmented ions with the higher masses were the same as observed in the full scan (i.e., from *m/z* 1122.4586 to 1069.4065). The observation of the first loss of SO_3_ (*m/z* 1042.5026, −1.3 ppm), in red in Figure 4, and the following fragmentation pattern of ammonium loss plus water losses (from *m/z* 1025.4769, −0.5 ppm, to 971.4436, −2.2 ppm) was consistent with the study of Estevez et al. [45] on gambierone, and with the fragmentations observed with the standard injected under the same conditions (Appendix A). The *m/z* 962.5446 (−2.7 ppm) was characteristic to sulfo-gambierone, as the ammonium adduct was still present despite the two −SO_3_ losses (probably in-source losses).

No fragments containing a sulfate group were observed because the fragmentation pathways for both sulfo-gambierones and for gambierone always started with the loss of sulfates followed by water losses before any other fragmentation. Hence, the position of both sulfate functions on sulfo-gambierone (marked with a green triangle on Figure 4) can only be hypothetical and based on the differences observed on the MS/MS spectra of gambierone and sulfo-gambierone.

Then, from *m/z* 945.5185 (−2.2 ppm) to 855.4614 (−7.5 ppm) the same ions as for gambierone ([45] and Appendix A) were observed. However, the ion characteristics to the right-hand side of gambierone (i.e., *m/z* 81.0699; 109.0648 and 219.1380) were not observed, suggesting that a difference between gambierone and sulfo-gambierone resides in this part of the molecule. The observation of *m/z* 121.0647 (−0.8 ppm) was suggested to result from the cleavage of the C38-C39 bond. Interestingly, the ion *m/z* 123.0798 (−5.2 ppm) was also observed and could correspond to the same cleavage as *m/z* 121.0647 if the carbonyl group on C40 was transformed in a hydroxyl one. This hypothesis implies the presence of a conjugated system of double bonds (in green in Figure 4) that would reduce the formation of the *m/z* 81.0699, and may suggest an in-source modification of the ion resulting from the sulfate loss and a subsequent rearrangement. The presence of the fragment *m/z* 163.0754 (+0.0 ppm) can correspond to the opening of the I-ring with either two hydroxyl groups and the double bond on C41-C42 or one hydroxyl group and one carbonyl group without the unsaturation on C41-42. The *m/z* 289.1811 (+4.4 ppm) was obtained with the opening of the H-cycle and was followed by one water loss at *m/z* 271.1712 (+7.2 ppm). As the molecular formula of sulfo-gambierone only differed by the number of sulfate groups, the number of unsaturated bonds was assumed to remain the same, and the presence of the double bond between C41 and C42 (Figure 4 in green) implied another modification elsewhere on the structure. No fragmentation was observed in gambierone on cycles A to F (Appendix A), and this observation suggested that either the structure promotes fragmentation on G-, H-, I-rings or prevents fragmentation on the A- to F-cycles. Our hypothesis of a modification of the C-cycle was deduced from the observed cluster of ions from *m/z* 639.3847 (−7.0 ppm) to 603.3649 (−5.2 ppm), which is absent for gambierone and could be the result of opening of the C-cycle, in which the double bond is absent compared to gambierone (Appendix A). This hypothesis was further supported by the observation of the cluster of ions from *m/z* 831.4507 (−2.2 ppm) to 777.4227 (+2.4 ppm) that matched with the left part of the molecule without a double bond on the C-cycle. The fragment *m/z* 543.3278 (−7.0 ppm) and the following water loss *m/z* 525.3197 (−2.6 ppm) agreed with the hypothesis of the presence of an extra double bond on the right part of the molecule (compared to gambierone) and may be favored by the opening of the C-cycle.

However, hypothesis of the location of the double bond on the C41–C42 of the molecule or on the location of the second sulfate group (green triangle in Figure 4) were made only based on the fragmentation pathway. Sulfate or water losses induce rearrangement in mass spectrometry, hence the presence of an unsaturation may be due as much to a difference in the molecular structure as to the result of fragmentation.

#### 2.2.4. Fragmentation Pathways of Dihydro-Sulfo-Gambierone

The targeted MS/MS spectra of the dihydro-sulfo-gambierone was also acquired and presented in Figure 5, and the proposed attribution of ion formula is presented in Appendix A.

The fragmentation pattern observed for dihydro-sulfo-gambierone was highly similar to the one of sulfo-gambierone, with a delta of 2 Da. The first water and sulfate losses from *m/z* 1124.4716 (−4.3 ppm) to *m/z* 857.4797 (−4.3 ppm) perfectly matched in term of intensity and ion ratios. Then, the cluster of ions from *m/z* 833.4647 (−4.2 ppm) to *m/z* 779.4365 (−3.8 ppm), resulting from fragmentation of the C40-C41 bond demonstrated that the difference of 2 Da was not located on the right-hand side of the molecule. The cluster of ions from *m/z* 755.4344 (−2.8 ppm) to *m/z* 719.4064 (−12.5 ppm) corresponding to the cleavage of the C38-C39 bond was observed in both gambierone (Appendix A) and dihydro-sulfo-gambierone. The difference in terms of exact mass of the measured ion (i.e., *m/z* 749.3892 for gambierone and *m/z* 755.4344 for dihydro-sulfo-gambierone) was consistent with the hypothesis of a difference of six hydrogens (2 unsaturations and one water loss) between the two compounds; hence we proposed the modification on the double bond of the I-cycle to explain the difference between sulfo-gambierone and dihydro-sulfo-gambierone. The proposed fragmentation pathway concerning the cluster from *m/z* 627.3852 (−6.2 ppm) to *m/z* 591.3635 (−7.6 ppm) was the opening of C- and D-cycles. This type of fragmentation is commonly observed for other polyethers [20], but was not found in either gambierone or sulfo-gambierone. The small fragments reported from *m/z* 289.1799 (−0.3 ppm) to *m/z* 121.0649 (+0.8 ppm) corresponded to the ones found in sulfo-gambierone based on our hypothesis (i.e., the double bond conjugated system that induces either a carbonyl or a hydroxyl for the right-hand side of the molecule). The ion *m/z* 93.0696 (−3.2 ppm) could be attributed to the cleavage between C39 and C40 after dehydration.

As no biosynthetic pathway is known for gambierone(s), the structures and fragmentation pathways proposed here are only putative and established (1) by comparison with the standard of gambierone, and (2) based on the high accuracy between the monoisotopic mass of the ion measured and the theoretical one.

Finally, the proposed hypotheses need to be confirmed by nuclear magnetic resonance (NMR) once sufficient amounts of those compounds are available at high purity. Obtaining such amounts is still a rather complicated challenge as *Gambierdiscus* spp. are slow-growing organisms, and cellular production of gambierone is comparatively low (<0.01–87 pg cell^−1^) [34], and a rather important amount of the compound is necessary due to its high molecular weight.

### 2.3. Toxicity of the New Analogues

The presence of a second sulfate group in the two molecules identified induced a higher polarity than gambierone and the resulting effect on toxicity was tested by Neuro2a cell-based assay. Viability after exposure to 556 ng mL^−1^ of the mix of sulfo-gambierones, gambierone, and 44-methylgambierone standards were compared to the negative methanol control (Figure 6) and showed no significant toxicity (i.e., 94–98, 95–95 and 93–93% viability, respectively). Details on sigmoid dose–response curves are provided in Appendix A.

Gambierone and 44-methylgambierone were suggested as weak sodium channel activators in previous studies [30,31] with a very low acute toxicity even by intraperitoneal injection [31], i.e., from 240 to 9,600 times less potent than ciguatoxins.

In this study, we confirmed the result of Boente-Juncal et al. [31], who reported no mortality of Neuro2a cells when exposed to gambierone at a concentration of 542 nM on the OV– condition. The two sulfated analogues did not induce mortality on Neuro2a cells at a concentration of 534 nM, supporting their absence of CTX-like activity (no significant difference between OV+ and OV− treatments). The presence of the second sulfate group does not modify the low activity of gambierone-like compounds on Neuro2a cells either with or without sensitization to sodium channel activators. The maximal concentrations used in this study were limited by the cost and availability of gambierone and 44-methylgambierone standards and of the mix of sulfo-gambierone analogues.

## 3. Materials and Methods

### 3.1. Chemicals

For the extraction process, acetone (AnalR NORMAPUR) and hexane (>95% J.T.Baker) were obtained from VWR (VWR Chemical, France).

LC-MS grade methanol and acetonitrile (Honeywell), formic acid (98% purity) and ammonium formate (10M) were purchased from Sigma Aldrich (Saint Quentin Fallavier, France). Water was deionized and purified at 18 MΩ cm thanks to a Milli-Q integral 3 system (Millipore, France).

For HRMS, acetonitrile, methanol, and high purity water (Optima LC–MS grade) were purchased from Fisher Chemical (Illkirch, France).

Standards of maitotoxin 1 were purchased from Wako (FUJIFILM Wako, Japan) and standards of gambierone and 44-methylgambierone were obtained from Cifga (Cifga laboratory, Spain)

For cell-based assay, the Neuro-2a cells line was obtained from ATCC (CCL-131^TM^, LGC standards, Molsheim, France). Roswell Park Memorial Institute 1640 medium (RPMI 1640) was obtained from Fisher Scientific (11554516, Fisher Scientific, France). Additives to the RPMI 1640 medium were fetal bovine serum (FBS) obtained from (1563397 Fisher Scientific, Illkirch, France), Penicillin, Streptomycin (LONZ17-603E, VWR, France) and 1 mM sodium pyruvate (L0642-100, VWR, France). Phosphate buffer saline without Ca and Mg was obtained from VWR (L0615-500, VWR, France). Accutase was purchased from VWR (L0950-100, VWR, France). For cell sensitization to sodium channel activators, Ouabain (O3125-1g, Sigma Aldrich, France) and Veratridin (ALEXBML-NA125-0010, VWR, France) were used. 3-(4,5-dimethylthiazol-2-yl)2,5-diphenyl tetrazolium bromide (MTT) was obtained from Sigma Aldrich (M5655-1g, Sigma Aldrich, Saint Quentin Fallavier, France).

### 3.2. Culture of Gambierdiscus Excentricus

The two *Gambierdiscus excentricus* strains (Bahamas Gam 5 and Pulley-Ridge Gam 2), initially isolated from the Bahamas and the Florida Keys, respectively, were both cultured in the Phycotoxin laboratory, Ifremer (Nantes, France) and in the Beaufort laboratory, NOAA (Beaufort, NC, USA). Details of the methodologies used are provided in Appendix A. All pellets were freeze dried and pooled for biomass accumulation (10.03 g, 124 million cells for compound isolation).

### 3.3. Cell Extraction

The lyophilized cell pellet was extracted, as presented in Appendix A, successively with 400 mL of acetone (kept separately), then 4 times with 400 mL of methanol, and finally 250 mL of aqueous methanol 80%. Each cycle of extraction was performed using vortex mixing for 30 s, ultrasonication for 15 min on ice (bath at 25 kHz), followed by another vortex mixing step for 30 s, and finally centrifugation at 4000× *g* for 4 min at 4 °C to separate the pellet from the supernatant. The acetone extract was concentrated to dryness. The methanol 80% extract was concentrated to 1.2 mL and the methanol 100% extract was concentrated to 40 mL.

### 3.4. Liquid-Liquid Partitioning

The concentrated 100% methanolic extract (40 mL) was partitioned twice with hexane (2 × 80 mL) to remove highly non-polar lipids (e.g., triglycerides). In a glass tube, the two layers were vortex mixed for 30 s every hour for 6 h and were then separated by centrifugation at 2000× *g* for 10 min at 4 °C. The two phases were conserved, concentrated under nitrogen down to 4.5 mL and stored at −20 °C.

### 3.5. Size Exclusion Fractionation

Fractionation of methanol 80% and methanol 100% extracts was performed as presented in the Appendix A. Size exclusion fractionation was carried out using Sephadex LH20 (GE Healthcare, VWR, France). The powder was swollen in 100% methanol overnight and then carefully packed into an open glass column (2 × 100 cm) and subsequently rinsed with methanol. Once packed, the total height of the stationary phase was 77 cm and the estimated column volume was 244 mL.

The MeOH 80% (1.2 mL) and MeOH 100% (2 × 2.2 mL) extracts were separately and successively fractionated on the same column, by eluting with MeOH 100% in isocratic mode and for each separation collecting 4 fractions of 10 mL followed by 120 fractions of 4 mL.

Fractions were analyzed by mass spectrometry with method 1, pooled according to the amount of gambierone analogs and concentrated down to 1.2 mL under nitrogen.

### 3.6. Reversed Phase Semi-Preparative Chromatography

Semi-preparative reversed phase chromatography was performed by injecting 900 µL of the pool of LH20 fractions containing the analogs of gambierone on an Uptisphere C18 column (250 × 10 mm, 5 µm, Interchim, France). The mobile phases were 100% MilliQ water (eluant A) and acetonitrile/water (95:5, *v*/*v*) (eluant B) and the flow rate was 4 mL min^−1^. The gradient used was as follows: 5% of B held during 5 min, then rose from 5% to 100% of B in 45 min, 100% of B held for 20 min and then from 100% to 70% of B in one min and held for 19 min. Fractions were collected each min from 0 to 60 min.

Fractions were analyzed by mass spectrometry using method 1, pooled according to the amount of gambierone analogs, concentrated under nitrogen without going to dryness and resuspended in methanol 90%.

### 3.7. Instrumental Conditions for Liquid Chromatography-LRMS: Screening of Compounds Produced by Gambierdiscus Species

Method 1: Screening of compounds produced by *Gambierdiscus* species was performed using a system composed of an ultra-high performance liquid chromatography (UHPLC) system (UFLC, Nexera, Shimadzu, Japan) coupled to a hybrid triple quadrupole-linear ion-trap mass spectrometer (API4000 QTRAP, Sciex, CA, USA) equipped with a TurboV source (ESI). The stationary phase was a Kinetex C18 column (50 × 2.1 mm, 2.6 µm 100 Å, Phenomenex, CA, USA) maintained at 40 °C. The mobile phases were 100% water (eluent A) and acetonitrile/water (95:5, *v*/*v*) (eluent B), both added with 2 mM ammonium formate and 50 mM formic acid. The flow rate was 0.4 mL min^−1^ and the elution gradient was as follows: 10 to 95% of B from 0 to 10 min, held at 95% B for 2 min and then back to the initial condition (10% B) at 12.1 min and held for 3 min. Analysis was performed using MRM in ESI^−^ mode. The parameters used were: curtain gas at 20 psi, turbo gas temperature at 400 °C, ion spray at −4500 V, gas 1 and 2 at 50 and 40 psi respectively, and the entrance potential was set at −10V. The transitions monitored were either based on the literature [27,29,35,44] or the result of optimization by infusion of standards (supplementary Appendix A). The instrument control, data processing, and analysis were conducted using Analyst software 1.7.2 (Sciex, CA, USA).

### 3.8. Instrumental Conditions for Liquid Chromatography-HRMS: Discovery and Characterisation of Gambierone Analogs

Analyses were carried out using a system composed of an UHPLC (1290 Infinity II, Agilent Technologies, CA, USA) coupled to a 6550 Ion Funnel Q-TOF (Agilent Technologies, CA, USA), equipped with a Dual Jet Stream^®^ ESI source. Two methods with different ionization and transmission parameters were used.

Method 2 was adapted from Sibat et al. [43] (able to detect a wide range of compounds): ESI positive and negative, gas temperature 160 °C, gas flow 11 L min^−1^, nebulizer 45 psi, sheath gas temperature 150 °C and sheath gas flow 11 L min^−1^, capillary voltage at +/−4500 V, nozzle voltage at 500 V, fragmentor at 365 V and ion funnel low pressure, high pressure and exit at 200 V, 100 V and 50 V, respectively.

Method 3 (optimized for gambierone and adapted from Yon et al. [20]): ESI positive only, gas temperature 150 °C, nebulizer 35 psi, sheath gas temperature 250 °C and sheath gas flow 6 L min^−1^, capillary voltage at +3500 V, nozzle voltage at 1000 V, fragmentor voltage at 135 V and ion funnel low pressure, high pressure and exit at 200 V, 200 V and 50 V, respectively.

The chromatography used a Kinetex C18 column (50 × 2.1 mm, 1.7 µm, 100 Å, Phenomenex, CA, USA) maintained at 40 °C. The mobile phases were 100% water (eluent A) and acetonitrile/water (95:5, *v*/*v*) (eluent B), both added with 2 mM ammonium formate and 50 mM formic acid. The flow rate was 0.4 mL min^−1^ and the elution gradient was as follows: 5% of B held for one minute, then from 5% to 100% of B in 10 min, held at 100% B for 2 min and then return to initial condition (5% B) at 13.5 min and held for 5.5 min.

Finally, method 4 combined the same ESI source parameters as method 3 but with a different chromatographic gradient allowing to separate the sulfated gambierone analogs. The elution gradient was as follows: 20% of B held for one minute, then from 20% to 100% of B in 13 min, held at 100% B for 3 min and then back to initial condition (20% B) at 17.5 min and held for 3.5 min.

For methods 2–4, the instrument was operated either in full scan at a scan rate of 2 spectra s^−1^ over a mass-to-charge ratio (*m/z*) from 100 to 1700 or in targeted MS/MS mode with a scan rate of 10 spectra s^−1^ for MS1 and 3 spectra s^−1^ for MS2 over a mass-to-charge ratio (*m/z*) from 50 to 1300. Reference mass *m/z* 922.0099 (hexakis phosphazene) was injected continuously at 1.5 mL min^−1^ with an isocratic pump over the entire run to ensure that no deviation in mass measurement occurred. Ammonium adducts of sulfo-gambierone and dihydro-sulfo-gambierone were the most intense ions observed in full scan mass spectra (Figure 3B,C) and consequently, targeted MS/MS spectra were acquired on the ammonium adduct to provide new elements on the possible structure of both compounds. The standard of gambierone was injected at a concentration of 20 µg mL^−1^. Instrument control and data treatment were carried out using the MassHunter software version B.08 (Agilent Technologies, CA, USA), and peaks were integrated using the algorithm Agile2.

### 3.9. Toxicity Assessment with Neuroblastoma Cell-Based Assay (CBA-N2a)

Neuroblastoma cell-based assay was adapted from Pisapia et al. [38]. Neuro2a cells were grown in a 75 cm² tissue-culture flask (734-0050, VWR, France) with RPMI-1640 medium containing 10% fetal bovine serum (FBS), 15 U L-1 Penicillin to 15 µg L-1 Streptomycin, and 1 mM Sodium pyruvate at 37 °C and 5% CO_2_ in a culture chamber (MCO-170AIC, Dominique Dutscher, Bernolsheim, France). Every two days, cells were rinsed with phosphate buffered saline (PBS) without Ca^2+^ and Mg^2+^, detached using accutase, and re-suspended in 10 mL RPMI-1640+10% FBS. Prior to conducting the assay, cells were counted (i.e., staining with Erythrosine-B and using a Malassez hemocytometer), then diluted in RPMI-1640+5% FBS (final concentration) and 200 µL were deposited into a 96 well plate (734-4058, VWR, France) to obtain 40,000 cells per well and incubated at 37 °C, 5% CO_2_ for 24 h to ensure a confluence >90%.

Cell exposure was performed the second day by adding either 10 µL of Ouabain/Veratridin (O/V^+^) at 77 µM and 7 µM respectively to assess the presence of sodium channel activators (standard of CTX3C and gambierones or sulfated analogs) or 10 µL of water (O/V^−^) to assess the presence of non-specific toxic compounds in samples. Then, portions (6 µL) of a serial dilution of either CTX3C standards (from 0.031–55.55 pg.mL^−1^ in the well), purified sulfo-gambierones (from 0.5 to 555.6 ng.mL^−1^ in the well) or gambierone and 44-methyl-gambierone (from 5.6 to 555.6 ng.mL^−1^ in the well) were added to each plate in triplicate.

Plates were homogenized and placed into the culture chamber for 20 h, then cell viability was assessed on the third day using a 3-(4,5-dimethylthiazol-2-yl)2,5-diphenyl tetrazolium bromide (MTT) colorimetric assay. Briefly, 50 µL of MTT in PBS was added at 0.8 mg mL^−1^ to each well and incubated for 30 min prior to measurement. The MTT solution was then aspirated, cells were disrupted with 100 µL of dimethyl sulfoxide (DMSO), homogenized, and the optical density of metabolized blue formazan was measured at 540 nm with a microplate reader (CLARIOstar PLUS, BMG Labtech, Champigny sur Marne, France). Sigmoidal dose-response curves were obtained with MARS software 4.00 (BMG Labtech, Champigny sur Marne, France), using a four-parameter logistic regression model to obtain the half-maximal effective concentration (EC_50_), with no blank correction.

## 4. Conclusions

*Gambierdiscus excentricus* was suggested as one of the most toxic *Gambierdiscus* species in the Atlantic Ocean by cell-based assays (on raw or partitioned extracts) [37,38,40]. In addition, this species has a characteristic toxin profile, since it is the only one to produce MTX4 [35]. Contrary to 16 other species of *Gambierdiscus* and 3 species of *Coolia* [33], and in accordance with Pisapia et al. [35], we did not detect gambierone or 44-methylgambierone in *G. excentricus* extract despite the large biomass analyzed. However, this study is the first report of two new analogs of gambierone, both having an additional sulfate group.

As reported by Longo et al. [46], 44-methylgambierone and gambierone can be detected either in the intracellular methanolic extract or in extracellular content with adsorption on HP20 resin. Hence, the high sensitivity in mass spectrometry detection for the two new sulfated analogs of gambierone discovered in this study may represent a useful tool for the identification of *Gambierdiscus excentricus*, either in extracts, culture media, or potentially in the field. Further studies are needed to ensure the specificity (i.e., production by other *Gambierdiscus* species) of these biomarkers.

The lack of toxicity of sulfo-gambierone and dihydro-sulfo-gambierone on Neuro2a cell-based assay demonstrated that these compounds were unlikely to be responsible for the high toxicity previously reported for *G. excentricus*. Further efforts are still required to determine the underlying compound(s), and hopefully will bring more knowledge on the production of polyethers by *Gambierdiscus* spp. in the Atlantic Ocean and Gulf of Mexico. Finally, this study confirmed the importance of using properly optimized high-resolution mass spectrometry to reduce misidentifications when working only with highly sensitive but low resolutive mass spectrometers.

## Figures and Tables

**Figure 1 marinedrugs-19-00657-f001:**
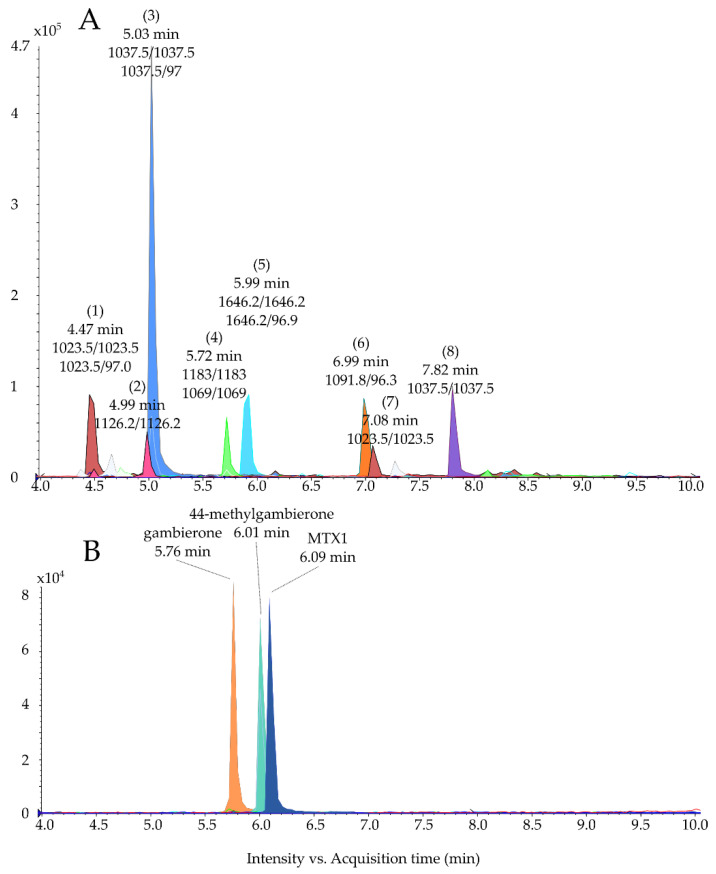
Method 1: (**A**) chromatogram of a methanol extract (250,000 cells mL^−1^) of *G. excentricus* (strain Bahamas Gam 5) using the multiple reaction monitoring (MRM) transitions in negative electrospray ionization (ESI^−^) mode presented in Appendix A and (**B**) chromatogram of a mixture of standards with gambierone (5.76 min) at 100 ng mL^−1^; 44-methylgambierone (6.01 min) at 100 ng mL^−1^ and maitotoxin 1 (MTX1) (6.09 min) at 2500 ng mL^−1^.

**Figure 2 marinedrugs-19-00657-f002:**
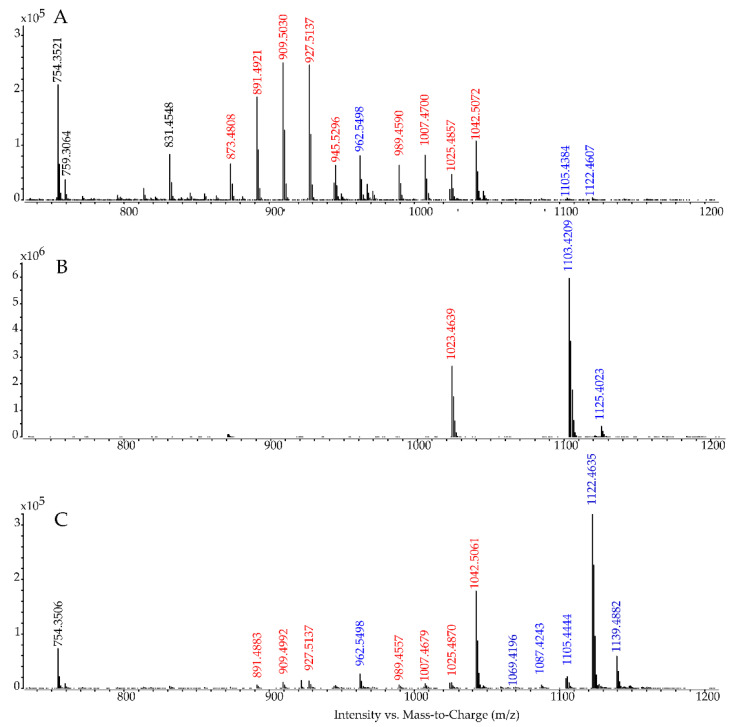
Zoom on the full scan of the putative gambierone analog (*m/z* 740-1200) acquired in ESI^+^ (**A**) and ESI^−^ (**B**) with method 2 (adapted from Sibat et al. [43]), and in ESI^+^ (**C**) with method 3 reducing in-source fragmentation and enhancing ionization and transmission of gambierone (adapted from Yon et al. [20]). The ions already reported for gambierone are shown in red, additional ions specific to the putative gambierone analog are shown in blue and ions not related to the compound in black.

**Figure 3 marinedrugs-19-00657-f003:**
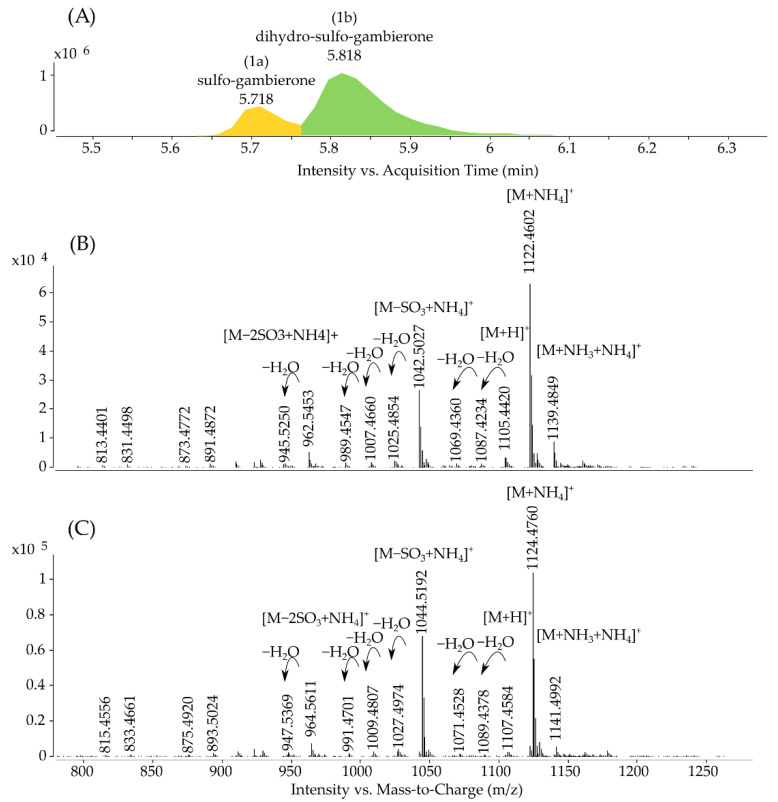
Extracted ion chromatogram (**A**) acquired with method 4 on the concentrated pool of preparative-HPLC fractions (29.8 µg gambierone eq. mL^−1^) with (**B**) resulting full scan spectra of the sulfo-gambierone (1a) and (**C**) resulting full scan spectra of the dihydro-sulfo-gambierone (1b) with their respective ion species.

**Figure 4 marinedrugs-19-00657-f004:**
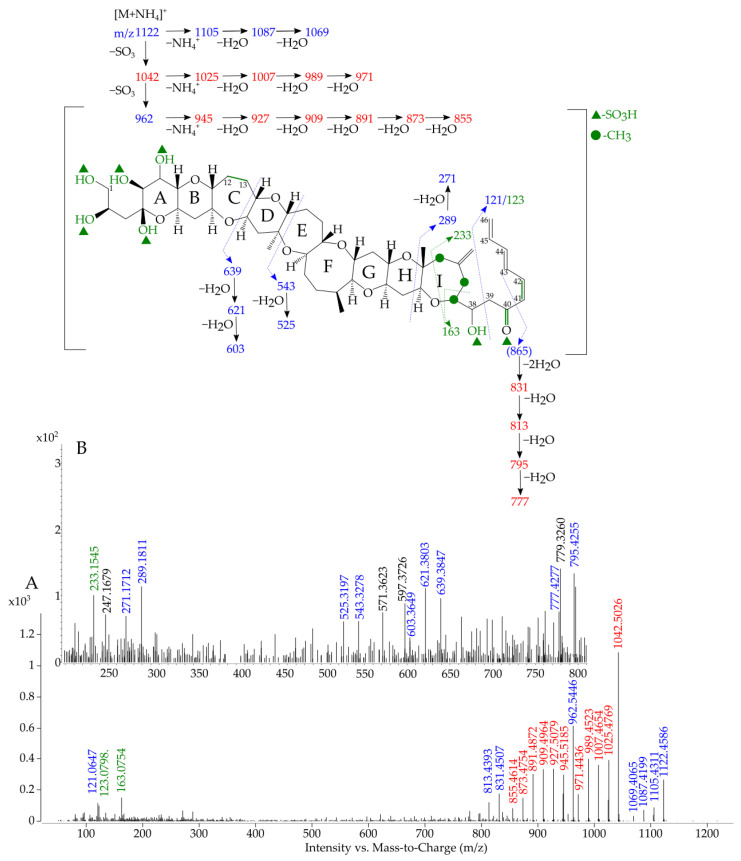
Targeted MS/MS spectrum of sulfo-gambierone (**A**) (average of collision energies 10,30 and 50 eV) acquired in ESI^+^ mode on the precursor *m/z* 1122.4608 ([M+NH_4_]^+^) and (**B**) zoom on the region *m/z* 200–800 at a collision energy of 30 eV. The mass-to-charge ratio (*m/z*) of spectral ions already reported in the literature for gambierone are shown in red, the proposed fragmentation pathway and corresponding ions are shown in blue and hypotheses and ions resulting from the modification of the structure compared to gambierone are shown in green. The assumptions on the position of the sulfate groups are marked by green triangles and on the position of the methyl group by green circles. * (Ions not detected).

**Figure 5 marinedrugs-19-00657-f005:**
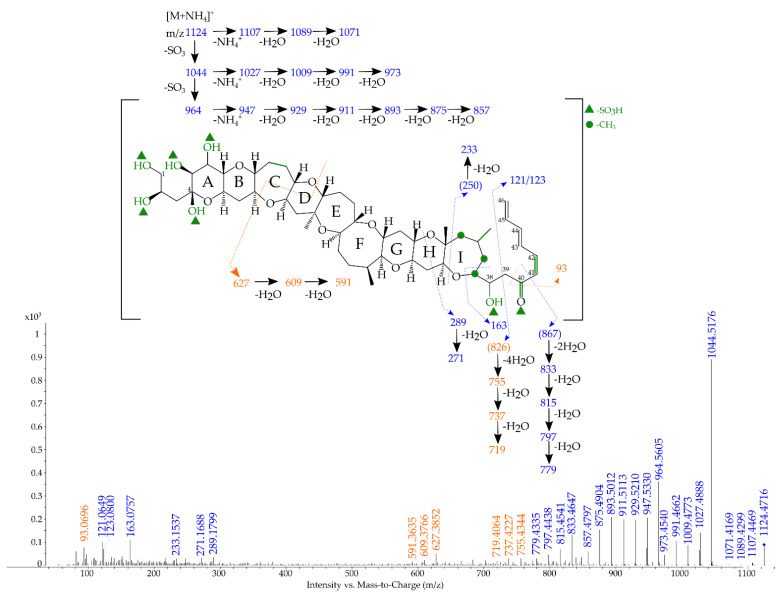
Targeted MS/MS fragmentation spectrum (average of collision energies 10, 30 and 50 eV) acquired in ESI^+^ mode on the precursor *m/z* 1124.4716 [M+NH_4_]^+^. The shared ions with a delta of 2 Da between sulfo-gambierone and dihydro-sulfo-gambierone are shown in blue. The proposal of the fragmentation pathway and corresponding ions specific to the new dihydro-sulfo-gambierone are shown in orange and the hypotheses on the modification of the structure between gambierone and dihydro-sulfo-gambierone and resulting ions are shown in green. The assumptions of the position of the sulfate groups are marked by green triangles and of the position of the methyl group by green circles. * (Ions not detected).

**Figure 6 marinedrugs-19-00657-f006:**
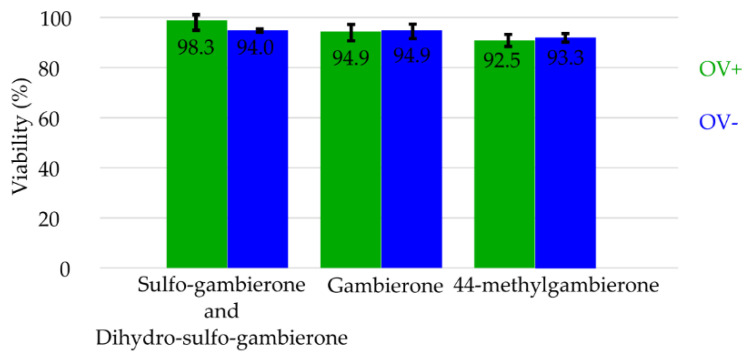
Viability of Neuro2a cells exposed for 20 h with purified sulfo-gambierones or gambierone and 44-methylgambietrone standards at a concentration of 556 ng mL^−1^ in the well, incubated with (OV+) and without (OV−) Ouabain and Veratridine.

**Table 1 marinedrugs-19-00657-t001:** Signals detected with low resolution mass spectrometry (LRMS) in ESI^−^ MRM mode using method 1 and confirmation status with high resolution mass spectrometry (HRMS) using method 2. Entries shown in blue are the corresponding signals reported in [35].

Low Resolution Mass Spectrometry (Method 1)			High Resolution Mass Spectrometry (Method 2)				
Chromatographic Peak	Retention Time (min)	MRM Transition	Retention Time (min)	Monoisotopic *m/z* Measured	*m/z* Theoretical	Δppm	Confirmation Status
(1)	4.47	1023.5/1023.5 1023.5/97.0	6.2	1023.4596	1023.4629 [30]	−3.2	Related
(2)	4.99	1126.2/1126.2	6.6	1126.6344	1125.5318 [41]	/	Not confirmed: Singly-charged ion
(3)	5.03	1037.5/1037.5 1037.5/97.0	6.72	1037.4599	1037.4780 [32]	−17.9	Putatively related Low mass accuracy
(4)	5.72	1183/1183 1069/1069	7.4	1181.8103	1183.6786 [27,44]	/	Not confirmed: Isotope M+3
(5)	5.99	1646.2/1646.2 1646.2/96.9	7.5	1645.2315	1645.2363 [41]	−2.9	Confirmed
(6)	6.99	1091.8/96.3	8.5	1091.5251	/		Not confirmed: Singly-charged ion
(7)	7.08	1023.5/1023.5	ND	ND			ND
(8)	7.82	1037.5/1037.5	9.5	1036.6185	1037.4780 [32]	/	Not confirmed: Isotope

**Table 2 marinedrugs-19-00657-t002:** Ion species corresponding to the accurate monoisotopic *m/z* of sulfo-gambierone acquired with method 2 and 3 in full scan mode. Mass differences (Δppm) were compared between theoretical exact mass and measured *m/z*. The ions already reported for gambierone are shown in red, additional ions specific to the putative gambierone analog are shown in blue.

Full scan ESI^+^ Method 2 (Δppm)	Full scan ESI^+^ Method 3 (Δppm)	Gambierone (C_51_H_76_O_19_S)		Sulfo-gambierone (C_51_H_76_O_22_S_2_)	
		Adduct Annotation	Theoretical Mono-Isotopic Mass (Da)	Adduct Annotation	Theoretical Mono-Isotopic Mass (Da)
	1139.4882 (+0.7)			[M+2NH_4_]^+^	1139.4874
1122.4607 (-0.1)	1122.4635 (+2.4)			[M+NH_4_]^+^	1122.4608
1105.4384 (-4.4)	1105.4444 (+9.2)			[M+H]^+^	1105.4342
	1087.4243 (+0.6)			[M-H_2_O+H]^+^	1087.4237
	1069.4196 (+6.1)			[M-2H_2_O+H]^+^	1069.4131
1042.5072 (+3.1)	1042.5061 (+2.0)	[M+NH_4_]^+^	1042.5040	[M-SO_3_+NH_4_]^+^	1042.5040
1025.4857 (+8.1)	1025.4870 (+9.4)	[M+H]^+^	1025.4774	[M-SO_3_+H]^+^	1025.4774
1007.4702 (+3.4)	1007.4679 (+1.1)	[M-H_2_O+H]^+^	1007.4668	[M-SO_3_-H_2_O+H]^+^	1007.4668
989.4590 (+2.7)	989.4557 (-0.6)	[M-2H_2_O+H]^+^	989.4563	[M-SO_3_-2H_2_O+H]^+^	989.4563
962.5498 (+2.7)	962.5477 (+0.5)			[M-2SO_3_+NH_4_]^+^	962.5472
945.5296 (+9.5)	945.5242 (+3.8)	[M-SO_3_+H]^+^	945.5206	[M-2SO_3_+H]^+^	945.5206
927.5137 (+4.0)	927.5102 (+0.2)	[M-SO_3_-H_2_O+H]^+^	927.5100	[M-2SO_3_-H_2_O+H]^+^	927.5100
909.5030 (+3.8)	909.4992 (-0.3)	[M-SO_3_-2H_2_O+H]^+^	909.4995	[M-2SO_3_-2H_2_O+H]^+^	909.4995
891.4921 (+3.6)	891.4883 (-0.7)	[M-SO_3_-3H_2_O+H]^+^	891.4889	[M-2SO_3_-3H_2_O+H]^+^	891.4889
873.4808 (+2.9)	873.4764 (-2.2)	[M-SO_3_-4H_2_O+H]^+^	873.4783	[M-2SO_3_-4H_2_O+H]^+^	873.4783

## Data Availability

Not applicable.

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
