# Peer review of "Sulfo-Gambierones, Two New Analogs of Gambierone Produced by Gambierdiscus excentricus"

_marinedrugs, 2021, doi:10.3390/md19120657_

Round 1

Reviewer 1 Report

Manuscript entitled „ Sulfo-Gambierones, Two New Analogs of Gambierone Produced by Gambierdiscus excentricus” is an interesting, well-written and well-planned experimental work, providing new data, concerning two new analogues of gambierone, both having an additional sulfate group, which can be considered as the useful tool for identification of Gambierdiscus excentricus. I fully support the publication of this manuscript; however I recommend the minor revision of manuscript. Small corrections should be made to the text as follows:

Introduction

line 56 – remove the before more likely

line 95 – should be species instead of spiece

Results

line 117 – explain in full name abbreviation ESI

line 122- remove (2017b)

line 135 – remove 2018

line 137 – explain abbreviation m/z and LRMS in full name

line 153 – which figure corresponds to (7)?

line 158 – remove (2020)

line 160 - put a parenthesis before Figure 1A and after (3)

line 195 – according to the instruction for authors:

All Figures, Schemes and Tables should be inserted into the main text close to their first citation and must be numbered following their number of appearance – so move Figure 2 from line 195 to line 210 before Table2

line 203 - put Figure 2A in parentheses and remove the commas

line 208 – remove (2020) and (2015)

line 223 - put Figure 2B in parentheses and remove the comma

line 228 – it is better to write (Figure 2A; i.e. from m/z 1025.4774 to m/z 945.5206)

line 237 - remove (2021)

line 240 – should be its instead their

line 289 - put parenthesis after Figure 4

line 305 – insert a space after on

line 311 – remove coma and add and after (Figure S9)

line 363 - explain in full name abbreviation NMR

line 384 - remove (2019)

Conclusions

line 398 – add Pisapia et al. before [32]

line 401 - remove (2019)

Figure 1 – according to the instructions for authors:

Acronyms/Abbreviations/Initialisms should be defined the first time they appear in each of three sections: the abstract; the main text; the first figure or table – write in full name abbreviations: MRM; ESI; MTX1

table 1 - write in full name abbreviations: LRMS; HRMS

Figure 2 – remove from description (2018) and (2021)

Materials and Methods

line 468 – put dot on the end of sentence

line 482 – use only abbreviation LRMS, remove full name because it was used earlier in main text

line 494-495 - use only abbreviation MRM and ESI-, remove full name because they were used earlier in main text

line 500 - use only abbreviation HRMS, remove full name because it was used earlier in main text

line 520 – put space between 5% B

line 524 - put space between 20% B

line 542 - write in full name abbreviation PBS

line 559 - write in full name abbreviation DMSO

References

line 605 – should be C.M.I; change the year of publication into 2021, add 102

line 611, 649, 652, 658, 661, 663, 672, 678, 687, 704, 707, 709, 712,  - the name of the journal should be written as an abbreviation

line 636 – position 14 - according to the rules of the journal, books should be described as follows:

Books and Book Chapters:

  1. Author 1, A.; Author 2, B. Book Title, 3rd ed.; Publisher: Publisher Location, Country, Year; pp. 154–196.
  2. Author 1, A.; Author 2, B. Title of the chapter. In Book Title, 2nd ed.; Editor 1, A., Editor 2, B., Eds.; Publisher: Publisher Location, Country, Year; Volume 3, pp. 154–196.

line 655 – in position 20 add number of pages

line 682, 687 – write the correct number of pages

line 705 – correctly write number of issue

Additionally, reference No 41 in not put into the main text of manuscript so remove it from the list of References or cite in main text

Author Response

Reviewer 1

Open Review

English language and style

( ) Extensive editing of English language and style required
( ) Moderate English changes required
(x) English language and style are fine/minor spell check required
( ) I don't feel qualified to judge about the English language and style

Yes

Can be improved

Must be improved

Not applicable

Does the introduction provide sufficient background and include all relevant references?

(x)

( )

( )

( )

Is the research design appropriate?

(x)

( )

( )

( )

Are the methods adequately described?

(x)

( )

( )

( )

Are the results clearly presented?

(x)

( )

( )

( )

Are the conclusions supported by the results?

(x)

( )

( )

( )

Comments and Suggestions for Authors

Manuscript entitled „ Sulfo-Gambierones, Two New Analogs of Gambierone Produced by Gambierdiscus excentricus” is an interesting, well-written and well-planned experimental work, providing new data, concerning two new analogues of gambierone, both having an additional sulfate group, which can be considered as the useful tool for identification of Gambierdiscus excentricus. I fully support the publication of this manuscript; however I recommend the minor revision of manuscript. Small corrections should be made to the text as follows:

We thank the reviewer for his overall positive comments and the detailed review of several aspects which allowed us to further improve the manuscript.

Please find below our point-by-point reply to the detailed comments.

Introduction

line 56 – remove the before more likely - Corrected as requested, see L56

line 95 – should be species instead of spiece - Corrected as requested, see L96

Results

line 117 – explain in full name abbreviation ESI – Corrected as requested L117. Now read “…in negative ionization electrospray (ESI-)….”

line 122- remove (2017b) – date was removed from line 122

line 135 – remove 2018 – removed from 1ine 136 as requested.

line 137 – explain abbreviation m/z and LRMS in full name – Lines 137-138 now read “The mass-to-charge ratio (m/z) values measured by low resolution mass spectrometry (LRMS) for each precursor ion…”

We thank the reviewer, abbreviations were explained and mistakes corrected in the revised manuscript.

line 153 – which figure corresponds to (7)?

We acknowledge the reviewer for his question. As the chromatographic peak (7) was only detected in LRMS (Figure 1 and Table 1) and not detected in HRMS we did not present this “absence of the compound” in supplementary material/figures. We added “data not shown”. Lines 153 – 154 of the revised manuscript, now read “…was not detected due to the difference of sensitivity of the HRMS for (7) (data not shown)”.

line 158 – remove (2020) – was removed as requested L161.

line 160 - put a parenthesis before Figure 1A and after (3) – parenthesis was added as requested.

line 195 – according to the instruction for authors:

All Figures, Schemes and Tables should be inserted into the main text close to their first citation and must be numbered following their number of appearance – so move Figure 2 from line 195 to line 210 before Table2 

We thank the reviewer for noticing, corrections were done in the revised manuscript.

line 203 - put Figure 2A in parentheses and remove the commas – Corrected as requested, see L197

line 208 – remove (2020) and (2015) - Corrected as requested, see L203

line 223 - put Figure 2B in parentheses and remove the comma - Corrected as requested, see L243

line 228 – it is better to write (Figure 2A; i.e. from m/z 1025.4774 to m/z 945.5206) - Corrected as requested, see L248

line 237 - remove (2021) - Removed as requested, see L257

line 240 – should be its instead their - Corrected as requested, see L260

line 289 - put parenthesis after Figure 4 - Corrected as requested, see L310-311

line 305 – insert a space after on - Corrected as requested, see L325

line 311 – remove coma and add and after (Figure S9) - Corrected as requested, see L332

line 363 - explain in full name abbreviation NMR – Revised L384-385, now read “… confirmed by nuclear magnetic resonance (NMR) once sufficient amounts of those com-pounds are available at high purity.”

line 384 - remove (2019) - Done L406.

Conclusions

line 398 – add Pisapia et al. before [32] Corrected as requested, see L420

line 401 - remove (2019) Corrected as requested, see L424

Figure 1 – according to the instructions for authors:

Acronyms/Abbreviations/Initialisms should be defined the first time they appear in each of three sections: the abstract; the main text; the first figure or table – write in full name abbreviations: MRM; ESI; MTX1

table 1 - write in full name abbreviations: LRMS; HRMS

Figure 2 – remove from description (2018) and (2021)

 We thank the reviewer for noticing these oversights. The acronyms included in the Figure 1, Table 1 and Figure 2 legends are now fully defined in the revised manuscript.

Materials and Methods

line 468 – put dot on the end of sentence - Corrected as requested, see L489

line 482 – use only abbreviation LRMS, remove full name because it was used earlier in main text - Corrected as requested, see L508

line 494-495 - use only abbreviation MRM and ESI-, remove full name because they were used earlier in main text - Corrected as requested, see L520

line 500 - use only abbreviation HRMS, remove full name because it was used earlier in main text - Corrected as requested, see L526

line 520 – put space between 5% B - Corrected as requested, see L546

line 524 - put space between 20% B - Corrected as requested, see L550

line 542 - write in full name abbreviation PBS – Corrected, now read “… phosphate buffered saline (PBS), see L571

line 559 - write in full name abbreviation DMSO – Corrected, now reads “… of dimethyl sulfoxide (DMSO)” see L589-590

References

line 605 – should be C.M.I; change the year of publication into 2021, add 102 - Corrected as requested, see L633-634

line 611, 649, 652, 658, 661, 663, 672, 678, 687, 704, 707, 709, 712,  - the name of the journal should be written as an abbreviation - Corrected as requested for reference No 3, 4, 9, 10, 12, 18, 22, 23, 24, 27, 38, 40 and 41.

line 636 – position 14 - according to the rules of the journal, books should be described as follows:

Books and Book Chapters:

  1. Author 1, A.; Author 2, B. Book Title, 3rd ed.; Publisher: Publisher Location, Country, Year; pp. 154–196.
  2. Author 1, A.; Author 2, B. Title of the chapter. In Book Title, 2nd ed.; Editor 1, A., Editor 2, B., Eds.; Publisher: Publisher Location, Country, Year; Volume 3, pp. 154–196.

We thank the reviewer and corrected accordingly in the revised manuscript.

line 655 – in position 20 add number of pages - Corrected as suggested, see L684

line 682, 687 – write the correct number of pages - Corrected as suggested, see L705 and L710

line 705 – correctly write number of issue - Corrected as suggested, see L728

Additionally, reference No 41 in not put into the main text of manuscript so remove it from the list of References or cite in main text.

Reference No 41. is now cited in Table 1

We thank the reviewer for the careful editing of the manuscript.

Reviewer 2 Report

Since the toxin content of the causative dinoflagellate is essential information for assessing the risk of seafood contaminated by that organism, such analysis is very useful for research and monitoring of seafood poisoning.  Therefore, it is very reasonable to publish this paper in the Journal. 

For one thing, the MS/MS experiments in negative ion mode reported by Naoki et al. (Rapid Commun. MS, 1997) show a clear pattern of cleavage of the ether ring at specific positions for these ladder-like polyether compounds.  

Author Response

Reviewer 2

Open Review

English language and style

( ) Extensive editing of English language and style required
( ) Moderate English changes required
( ) English language and style are fine/minor spell check required
(x) I don't feel qualified to judge about the English language and style

Yes

Can be improved

Must be improved

Not applicable

Does the introduction provide sufficient background and include all relevant references?

(x)

( )

( )

( )

Is the research design appropriate?

( )

(x)

( )

( )

Are the methods adequately described?

(x)

( )

( )

( )

Are the results clearly presented?

( )

(x)

( )

( )

Are the conclusions supported by the results?

(x)

( )

( )

( )

Comments and Suggestions for Authors

Since the toxin content of the causative dinoflagellate is essential information for assessing the risk of seafood contaminated by that organism, such analysis is very useful for research and monitoring of seafood poisoning.  Therefore, it is very reasonable to publish this paper in the Journal. 

For one thing, the MS/MS experiments in negative ion mode reported by Naoki et al. (Rapid Commun. MS, 1997) show a clear pattern of cleavage of the ether ring at specific positions for these ladder-like polyether compounds.  

We acknowledge the reviewer for pointing out these earlier results. However, we did not perform negative ionization MS/MS as the spectra tend to have few revelatory fragment ions due to the prominent sulfate loss. We also do not have access to a high energy collision cell that allows systematic fragmentation of the ether rings, unlike Naoki and colleagues.

Submission Date

27 October 2021

Date of this review

09 Nov 2021 06:00:37

Reviewer 3 Report

General

The manuscript presents some advancements and contributions to the phenomenon of Ciguatera Fish Poisoning, which still possesses practical challenges due to the limitations to grow the producing algae under controlled conditions and the lack of availability of purified standards. The authors have attempted to describe two new analogues, with some limitations, which renders interest for continuing research in this topic. For a complete evidence of toxic properties of the new analogues, I would recommend performing mouse bioassays in comparison with the cell assays presented by the authors in this manuscript. I am aware of the availability limitation of ciguatoxins in the market, but it is something that the authors may want to consider for future research.

Particular

Line 16. Change to “fish contaminated with ciguatoxins…”

Line 17. Correct typo to “Gambierdiscus”

Line 23. Change to “a purified extract from 124 million cells…”

Lines 31-32. How specific are the two new sulphated analogues described by the authors? Did the authors also use other Gambierdiscus or Fukuyoa species to determine that only G. excentricus produces these analogues and hence use them as biomarkers for this particular species? From what it’s mentioned in the Introduction, it appears that MTX4 seems to be a better biomarker for G. excentricus.

Line 40. It is recommended that the authors use the commonly and widely known acronym CFP (ciguatera fish poisoning) throughout the manuscript.

Line 86. “only” is repeated, use it only once.

Figure 2. Legend/title in the y axis is missing, or it is recommended to label each axis independently.

Line 371. Replace “exposition” with “exposure”

Line 398. Mention the reference (author, year), not only the number

Lines 398 & 403-405. This sounds contradictory. First the authors mentioned that they did not detect gambierone or 44-methylgambierone in G. excentricus (line 398), and then they mentioned that those two analogs can be used for G. excentricus identification. This needs to be clarified and reworded.

Line 440. Make sure and check throughout the manuscript that “Gambierdiscus excentricus” is always in italics.

Line 497-498. It is more ethical to include the author and year of the method, followed by the reference number [32].

Lines 539-564. Critically, the reference of this method is missing.

Lines 593-594. The first sentence can be omitted and start with: Conceptualization…

Author Response

Reviewer 3

Open Review

English language and style

( ) Extensive editing of English language and style required
( ) Moderate English changes required
(x) English language and style are fine/minor spell check required
( ) I don't feel qualified to judge about the English language and style

Yes

Can be improved

Must be improved

Not applicable

Does the introduction provide sufficient background and include all relevant references?

( )

(x)

( )

( )

Is the research design appropriate?

( )

(x)

( )

( )

Are the methods adequately described?

( )

(x)

( )

( )

Are the results clearly presented?

(x)

( )

( )

( )

Are the conclusions supported by the results?

( )

(x)

( )

( )

Comments and Suggestions for Authors

General

The manuscript presents some advancements and contributions to the phenomenon of Ciguatera Fish Poisoning, which still possesses practical challenges due to the limitations to grow the producing algae under controlled conditions and the lack of availability of purified standards. The authors have attempted to describe two new analogues, with some limitations, which renders interest for continuing research in this topic. For a complete evidence of toxic properties of the new analogues, I would recommend performing mouse bioassays in comparison with the cell assays presented by the authors in this manuscript. I am aware of the availability limitation of ciguatoxins in the market, but it is something that the authors may want to consider for future research.

We thank the reviewer for his comments and the detailed review of several aspects which allowed us to further improve the manuscript. We will consider a collaboration to run mouse bioassays in future research.

Please find below our point-by-point reply to the detailed comments.

Particular

Line 16. Change to “fish contaminated with ciguatoxins…” - Corrected as requested, see L16

Line 17. Correct typo to “Gambierdiscus” - Corrected as requested, see L17

Line 23. Change to “a purified extract from 124 million cells…” - Corrected as requested, see L23

Lines 31-32. How specific are the two new sulphated analogues described by the authors? Did the authors also use other Gambierdiscus or Fukuyoa species to determine that only G. excentricus produces these analogues and hence use them as biomarkers for this particular species? From what it’s mentioned in the Introduction, it appears that MTX4 seems to be a better biomarker for G. excentricus.

We thank the reviewer to point out that no data are provided in the manuscript concerning the specificity of the two sulfated analogs of gambierone as biomarkers of G. excentricus only. We thus clarified to read “…Further studies are needed to ensure the specificity (i.e. production by other Gambierdiscus species) of these biomarkers..” L428-429. We would like to mention that a larger study is underway to answer this question.

Line 40. It is recommended that the authors use the commonly and widely known acronym CFP (ciguatera fish poisoning) throughout the manuscript.

We acknowledge the reviewer for his suggestion but we preferred to not limit the ciguatera poisoning to fish, as ciguatoxins were detected in shellfish too (e.g. Roué et al., 2018). We used Ciguatera food poisoning instead (CFP).

­Line 86. “only” is repeated, use it only once. - Corrected as requested, see L86

Figure 2. Legend/title in the y axis is missing, or it is recommended to label each axis independently.

The label of the X-axis contains both the titles of y-axis and x-axis by denoting intensity versus mass-to-charge ratio. This is customary in chromatography and mass spectrometry and has been previously accepted in the Marine Drugs journal (e.g. Estevez et al., 2021)

Line 371. Replace “exposition” with “exposure” - Corrected as requested, see L393

Line 398. Mention the reference (author, year), not only the number We agree with the reviewer. However, we finally added more references here to better support our statement. Therefore we did not add the names of all authors of the references. ( L417)

Lines 398 & 403-405. This sounds contradictory. First the authors mentioned that they did not detect gambierone or 44-methylgambierone in G. excentricus (line 398), and then they mentioned that those two analogs can be used for G. excentricus identification. This needs to be clarified and reworded.

We thank the reviewer for pointing the potentially contradictory statement.  In response, we modified the sentence Lines L22-L32 to explain that only the two new analogs of gambierone can be used as biomarkers of G. excentricus. In addition L419-L430 will now read as follows,

            “…Contrary to 16 other species of Gambierdiscus and 3 species of Coolia [30], and in accordance with Pisapia et al. [32], we did not detect gambierone or 44-methylgambierone in G. excentricus extract despite the large biomass analyzed. However, this study is the first report of two new analogs of gambierone, both having an additional sulfate group….

And

            … Hence, the high sensitivity in mass spectrometry detection for the two new sulfated analogs of gambierone discovered in this study, may represent a useful tool for the identifica-tion of Gambierdiscus excentricus, either in extracts, culture media or potentially in the field. Further studies are needed to ensure the specificity (i.e. production by other Gambierdiscus species) of these biomarkers..”

Line 440. Make sure and check throughout the manuscript that “Gambierdiscus excentricus” is always in italics. Now consistently italicized throughout the manuscript.

Line 497-498. It is more ethical to include the author and year of the method, followed by the reference number [32].

We agree with the reviewer. However, we finally added more references here. Therefore we did not add the names of all authors of the references. See line L523.

Lines 539-564. Critically, the reference of this method is missing.  Corrected, now reads “… Neuroblastoma cell-based assay was adapted from Pisapia et al., [35]….” see L567

Lines 593-594. The first sentence can be omitted and start with: Conceptualization…

We agree with the reviewer, the first sentence was removed in the revised manuscript

Submission Date

27 October 2021

Date of this review

09 Nov 2021 22:48:02

Reviewer 4 Report

The manuscript is well written and organized, adding more analogues to those already known from the gambierone family.

I have only minor comments:

Line 86 - «were only reported in only…» please avoid the tautology

lines 110-113 – Objectives: please refine the location of strain isolation. The use of ‘Atlantic area’ in the manuscript is rather vague, as there is more than one endemic area in the Atlantic, such as the Caribbean or the Macaronesia region, located in opposite margins of the Atlantic Ocean (also line 411).

Line 429- the «RoswellParkMemorialInstitute1640(RPMI1640)», is a cell line or a culture medium?

Lines 441- the «Phycotoxin laboratory» belongs to which Institute?

Line 441- I cannot find the abbreviation for the «CCHFR» laboratory in the author’s addresses.

Line 529- no details on how hexakis phosphazene was injected into the system during the run: another pump?, mixed with mobile phase?

Line 531- «Ammonium adducts of both analogs» this expression is rather vague as there were several standards and novel compounds analyzed. Please improve.

Line 535- the software used was described here in detail. No such details were provided for the low-resolution MS system described in section 4.8. Please uniformize.

Author Response

Reviewer 4

Open Review

English language and style

( ) Extensive editing of English language and style required
( ) Moderate English changes required
(x) English language and style are fine/minor spell check required
( ) I don't feel qualified to judge about the English language and style

Yes

Can be improved

Must be improved

Not applicable

Does the introduction provide sufficient background and include all relevant references?

(x)

( )

( )

( )

Is the research design appropriate?

(x)

( )

( )

( )

Are the methods adequately described?

( )

(x)

( )

( )

Are the results clearly presented?

(x)

( )

( )

( )

Are the conclusions supported by the results?

(x)

( )

( )

( )

Comments and Suggestions for Authors

The manuscript is well written and organized, adding more analogues to those already known from the gambierone family.

We thank the reviewer for his positive comments and the detailed review of several aspects which allowed us to further improve the manuscript.

Please find below our point-by-point reply to the detailed comments.

I have only minor comments:

Line 86 - «were only reported in only…» please avoid the tautology

We thank the reviewer to point out the mistake, that was corrected in the revised manuscript L86

lines 110-113 – Objectives: please refine the location of strain isolation. The use of ‘Atlantic area’ in the manuscript is rather vague, as there is more than one endemic area in the Atlantic, such as the Caribbean or the Macaronesia region, located in opposite margins of the Atlantic Ocean (also line 411).

We corrected the sentence in the materials & methods section to read as follows to clarify both the origin of strains and the place of culture: “The two Gambierdiscus excentricus strains (Bahamas Gam 5 and Pulley-ridge Gam 2), initially isolated from the Bahamas and the Florida Keys, respectively, were  both cultured in the Phycotoxin laboratory, Ifremer (Nantes, France) and in the Beaufort laboratory, NOAA (Beaufort, NC, USA)”.

Line 429- the «RoswellParkMemorialInstitute1640(RPMI1640)», is a cell line or a culture medium?

We thank the reviewer to point out that this part was not clear enough in the manuscript. We did add “medium” L453.

Lines 441- the «Phycotoxin laboratory» belongs to which Institute?

We thank the reviewer to point out that this part was not clear enough in the manuscript. We did add “Ifremer (France)” L466

Line 441- I cannot find the abbreviation for the «CCHFR» laboratory in the author’s addresses.

We acknowledge the reviewer to point out that it was not clear, we did remove this abbreviation in the revised manuscript.

Line 529- no details on how hexakis phosphazene was injected into the system during the run: another pump?, mixed with mobile phase?

We thank the reviewer to point out that this part was not clear enough in the manuscript. We did add “at 1.5 mL min-1 with an isocratic pump” L557

Line 531- «Ammonium adducts of both analogs» this expression is rather vague as there were several standards and novel compounds analyzed. Please improve.

We thank the reviewer to point out that this part was not clear enough in the manuscript. We did modified the sentence as follows: “Ammonium adducts of sulfo-gambierone and dihydro-sulfo-gambierone were…” L558

Line 535- the software used was described here in detail. No such details were provided for the low-resolution MS system described in section 4.8. Please uniformize.

We agree with the reviewer, we added “ The instrument control, data processing and analysis were conducted using Analyst software 1.7.2 (Sciex, CA, USA)” to uniformize the revised manuscript. L524-525

Submission Date

27 October 2021

Date of this review

09 Nov 2021 16:21:10